# Comparative Evaluation of Chemical Composition and Nutritional Characteristics in Various Quinoa Sprout Varieties: The Superiority of 24-Hour Germination

**DOI:** 10.3390/foods13162513

**Published:** 2024-08-12

**Authors:** Beier Cao, Changjian Bao, Zhiqiang Zhu, Yanning Gong, Junyu Wei, Zhenguo Shen, Nana Su

**Affiliations:** College of Life Sciences, Nanjing Agricultural University, Nanjing 210095, China; 2022116002@stu.njau.edu.cn (B.C.); 2022216032@stu.njau.edu.cn (C.B.); 2023816121@stu.njau.edu.cn (Z.Z.); 2023116018@stu.njau.edu.cn (Y.G.); 2017816131@stu.njau.edu.cn (J.W.); zgshen@njau.edu.cn (Z.S.)

**Keywords:** quinoa sprout, nutrition quality, saponins

## Abstract

Quinoa (*Chenopodium quinoa* Willd) sprouts are rich in bioactive compounds that offer numerous health benefits. However, limited research exists on their cultivation, nutritional value, and processing potential. This study compared the nutritional composition and antioxidant activity of quinoa sprouts from different varieties at various time points. Results showed a general increase in most nutrients over time. At the 24 h mark, JQ-W3 exhibited a 17.77% increase in leucine, 1.68 times higher than in eggs, along with a 6.11-fold elevation in GABA content. JQ-B1 exhibited the preeminent antioxidant potency composite (APC) score. Saponins, known for their bitter taste, decreased at 12 h but returned to original levels by 24 h. Based on nutritional components and saponin content, 24 h sprouted black quinoa JQ-B1 and white quinoa JQ-W3 were selected, providing a basis for quinoa sprout development in the food industry. These findings contribute to the understanding and utilization of quinoa sprouts.

## 1. Introduction

Quinoa (*Chenopodium quinoa* Willd) is a pseudo-cereal of the Amaranthaceae family [1]. More and more evidence showed that quinoa is a kind of whole grain, whose material content has a similarity with the requirements of the normal human body’s intake, and whose seeds contain a large number of phytochemicals [2]. Studies have found that quinoa contains a large number of proteins and biologically available essential amino acids, unsaturated lipids, complex carbohydrates, dietary fiber, and other useful bioactive compounds, including polyphenolic compounds, such as phenolic acids, flavonoids, lignin, and tannins [3]. The main phenolic acids in its seeds are ferulic acid, caffeic acid, coumarin and benzoic acid, and the main flavonoids of horseradish, myricetin, and quercetin [4]. In addition, studies have shown that polyphenol derivatives, as the important plant nutrient, have strong antioxidant activities, which can inhibit the activity of some pathogenic organisms in human body and thus benefit human health [2].

A quinoa sprout is a kind of edible vegetable formed after the germination of quinoa seeds. Due to the scarcity of various resources and the limitation of cultivation technology, the quinoa sprout industry is still in its infancy and has not been widely used in production. The existing literature shows that the nutritional value of quinoa sprouts is even higher than that of quinoa seeds and, after germination, the sugars, essential amino acids, mineral elements, vitamins, and other nutrients in the quinoa sprouts are significantly increased [5,6]. Studies have found that quinoa sprouts contain lysine, which is higher than that of other ordinary grains [7]. Because lysine cannot be synthesized in the human body and can only be ingested through food, quinoa seedlings are expected to become a healthy food to supplement lysine in the human body. Quinoa sprouts are rich in vitamin B [8], which plays an important role in protecting human nervous tissue and maintaining the functions of the heart and the nervous system. Therefore, quinoa sprouts have potential value for the development of advanced healthy food. However, at present, the related research on quinoa planting is mostly focused on improving the yield and stress resistance of quinoa, while the related research on quinoa composition is mostly focused on the nutritional components of quinoa and the analysis of its toxic components, and the existing research on quinoa seeds is mostly focused on the discussion and analysis of the edible value and processing application of [9,10,11], but the quinoa sprouts’ germination, nutritional value, processing, and utilization have not been fully studied and discussed. Therefore, quinoa sprouts have a strong development potential and are excellent materials for studying the content changes of pseudo-cereal seeds after germination. Thus, this paper will study the changes in the various nutritional components, phenolic compounds, saponin, vitamins, and antioxidant activities during the growth and development of quinoa sprouts.

In this study, the quinoa varieties suitable for industrial production were initially selected through cooperation with the enterprises. This paper further discussed the changes in the trends of important nutritional indicators to visualize the nutritional value of quinoa sprouts. This study can provide references for the large-scale industrialization and standardized production of quinoa sprouts in the future and provide a theoretical basis for the healthy food option of quinoa sprouts, which has good market development prospects and application value.

## 2. Materials and Methods

### 2.1. Material

Eight varieties of quinoa were selected as the experimental materials, which were provided by Jiaqi Agricultural Science and Technology Co., Ltd. Taiyuan, China. Common names of each variety and information of origins are shown in Table 1.

To prepare the quinoa sprouts, the seeds were first soaked in 1% NaClO (*v*/*v*) for 5 min. Further, the seeds were steeped in deionized water for 4 h at 25 °C, after which the seeds were spread in a thin layer on a tray which was covered with a moist muslin cloth. The trays were placed in a growth chamber (Safe, CRY-500, Hangzhou, China) and the seeds were allowed to germinate for 0, 12, 24, and 36 h at 24 °C and 60% humidity, under dark conditions. The samples were quickly placed into liquid nitrogen and stored at −80 °C.

### 2.2. The Growth Index Measurement

One thousand seeds were drawn at random from the bulk sample to determine the seed diameter, hypocotyl length, and 1000-sprout weight of the quinoa sprouts (0, 12, 24, and 36 h). A digital micrometer (with a minimum count of 0.01 mm) was used to measure the seed diameter and hypocotyl length. The weight of 1000 sprouts were measured with a balance of 1/10,000 sensitivity (Precisa, XB220A, Dietikon, Switzerland).

### 2.3. The Soluble Sugar, Soluble Protein, and Free Amino Acid Content Measurement

The quantification of soluble sugars, soluble proteins, and free amino acids in the quinoa sprouts was performed using a Synergy HT MultiMode Microplate Reader (Biotek, Rochester, VT, USA).

For the total soluble sugar assay, the content was analyzed by the anthrone colorimetry method [12]. In brief, powdered samples were immersed in 5 mL of deionized water, heated at 100 °C for 1 h, and then centrifuged at 4000 g for 20 min. To 2 mL of the supernatant (extraction), 3 mL of anthrone solution (0.4% in concentrated H_2_SO_4_) was then added. The reaction was stopped after 20 min at 90 °C, the soluble sugar content was quantified by measuring the absorbance at 620 nm, and the results were compared with a glucose standard solution (0–200 μL/mL).

To perform the soluble protein assay, 0.2 g of quinoa powder was dissolved in 5 mL of deionized water, centrifuged at 10 °C and 3000 rpm for 10 min. Next, 10 μL of the supernatant was mixed with 190 μL of Coomassie Brilliant Blue G-250 solution for 2 min. Subsequently, the absorbance of all samples was measured at 595 nm. The identical procedure was repeated for the standard solution of bovine serum albumin, and a calibration curve was generated.

The content of free amino acids was determined by mixing 1 mL of the extraction solution with 0.5 mL of 3% ninhydrin solution, followed by boiling in a water bath for 100 min. After cooling to room temperature, 5 mL of 95% ethanol was added, and a standard curve was prepared using standard amino acid solutions (dissolve 46.8 mg dried leucine in 10% isopropanol, make up to 100 mL, then withdraw 5 mL and dilute to 50 mL with water for a 5 μg/mL amino nitrogen standard solution) following the same method, and 200 μL of the mixture was taken for analysis at 570 nm.

### 2.4. The Amino Acid and GABA Content Measurement

The determination of amino acid and GABA content was assessed using the L-8900 automatic amino acid analyzer through HPLC technology [13]. First, 0.1 g of dry sample powder was added to 5 mL 6 M hydrochloric acid at 110 °C for 24 h. This was adjusted to 100 mL with double-distilled water, after which the solution underwent rotary evaporation, reconstitution with 1 mL water, and filtration through a 0.22 μm filter. Amino acid contents were determined based on standard peak time and area. The analysis column (4.6 mm × 60 mm, 3 μm) and reaction column were maintained at temperatures of 57 and 135 °C, respectively. Pump A (citrate buffer) and pump B (ninhydrin buffer) operated at flow rates of 0.40 and 0.35 mL/min, respectively. GABA detection was conducted at 570 nm using a dual-channel UV detector.

### 2.5. The Vitamin Content Measurement

Vitamin content was determined using HPLC [14], focusing on VB1 (Thiamine), VB2 (Riboflavin), VB6 (Pyridoxine), and VC (Ascorbic Acid). Powdered samples were extracted with 2% metaphosphate for 30 min. Supernatants were analyzed by HPLC (e2695 pump, 2998PDA detector, Waters, Milford, MA, United States) with a C18 column (4.6 × 250 mm) at 30–40 °C, 0.8 mL/min flow rate, and 10 µL injection volume. Eluent A (2% metaphosphate acid) and eluent B (methanol) were used. Vitamins were detected at 254 nm. The standard curve was used for vitamin content calculation.

### 2.6. The Total Phenol Content (TPC) and Total Flavonoid Content (TFC) Measurement

Total phenol content was quantified using the Folin phenol colorimetric method [15]. First, 0.1 g of the dried quinoa samples were extracted with 6 mL solution consisting of 70% acetone, 29.5% deionized water, and 0.5% acetic acid (*v*/*v*/*v*) through a 30 min sonication at 20 °C (extraction liquids encompass both total flavonoids and total phenols). Supernatant is then mixed with Folin reagent and 7.5% Na_2_CO_3_ and incubated for 30 min in the dark. The same operation was carried out with different concentration gallic acid standard solution as the control. Absorbance was then measured at 765 nm for phenol quantification. A calibration curve with gallotannin content against absorbance values was created, and the calculations were performed using the following formula:Total phenol content (mg GAE/g DW) = (x × V)/m(1)

In the formula, x is the total phenol content obtained from the standard curve (mg/mL); V is the total volume of the sample extract (mL); and m is the weight of the sample (g).

The total flavonoid content was quantified using a modified colorimetric method [16]. A diluted flavonoid solution (1 mL) was combined with 80% (*v/v*) ethanol (5 mL) and 5% (*w/w*) NaNO_2_ (1 mL) for 6 min. Afterwards, 10% AlCl_3_ (*w/w*) (1 mL) was added and mixed, followed by the addition of 1 mol/L NaOH (10 mL) after 6 min. The same procedure was performed using rutin standard solutions of varying concentrations as controls. Following a 15 min incubation period, the absorbance at 510 nm was measured against a sample-free blank. A standard curve was plotted, with rutin content on the x-axis and absorbance values on the y-axis. The total flavonoid content was calculated using the following formula:Total flavonoid content (mg Rutin/100 g DW) = (x × V)/m (2)

In the formula, x is the total flavonoid content obtained from the standard curve (mg/mL); V is the total volume of the sample extract (mL); and m is the weight of the sample (g).

### 2.7. The Saponin Content Measurement

The saponins were extracted from the sample using a modified method of Fenwick (1983) and Sharma (2022) [17,18]. A dehydrated sample (0.5 g) was extracted with 20 mL of acetone for at least 24 h to remove the lipids, pigments, and impurities. Then, 5 mL of methanol was used as a replacement solvent, and the extraction continued for an additional 24 h. Afterward, 1 mL of the methanolic extract was heated to evaporate the methanol in a boiling water bath. Following cooling, 2 mL of ethyl acetate, 1 mL of a reagent composed of 0.5 mL anisaldehyde and 99.5 mL ethyl acetate, and 1 mL of concentrated sulfuric acid were added. The mixture was thoroughly mixed and allowed to stand for 10 min at room temperature. Absorbance was measured at 430 nm, and the saponin content was expressed as mg per g of dry sample (mg saponins/g), using saponin (extrapure) as the standard reference.

### 2.8. Antioxidant Activities Measurement

DPPH and FRAP assays were performed according to the protocol by Rufino et al. (2011) [19]. Samples (0.1 g) were homogenized with 5 mL 70% ethanol, sonicated for 10 min, and centrifuged at 11,000 rpm for 10 min at 20 °C to obtain the extraction solution.

DPPH radical scavenging capacity was assessed by adding 100 µL of the extraction solution to a 0.1 mM DPPH–methanol solution. After 30 min of dark incubation, the absorbance at 517 nm was measured using an ethanol blank. The DPPH radical scavenging activity was quantified as the percentage inhibition using the following equation:DPPH scavenging activity (%) = (1 − Asample/Acontrol) × 100(3)

In the FRAP assay, the extraction solution (100 μL) was combined with a reaction mixture consisting of 0.3 M acetate buffer, 10 μM TPTZ, and 20 μM FeCl_3_ (3 mL). Following a 20 min incubation period, the absorbance at 593 nm was measured. The sample absorbance values were compared to a calibration curve constructed using various FeSO_4_ concentrations to determine the FRAP.

ABTS radical scavenging was determined using a total antioxidant capacity assay kit (Jiancheng Bioengineering Institute, Xi’an, China). The tissue was accurately weighed in a 1:9 ratio (g:mL). Then, 9× saline was added, the mixture was homogenized and centrifuged, and the supernatant was collected. Up to 10 μL of the sample was mixed with 20 μL 10× assay buffer and 170 μL ABTS solution. After incubating for 6 min, the absorbance was measured at 405 nm (Biotek, Synergy HT MultiMode Microplate Reader, Winooski, VT, USA), and then calculated with the Trolox standard solution. The APC (antioxidant potency composite) index was computed for each antioxidant method using the following equation:Antioxidant index score = [(sample score/best score) × 100](4)

The APC index was determined by averaging the antioxidant index scores of each method.

### 2.9. Statistical Analysis

The raw data were statistically analyzed using SPSS 19.0, while data visualization was accomplished with Origin 2021, R Studio 4.1.0, and Adobe Illustrator 2022. Results are expressed as mean ± standard deviation (SD), derived from a minimum of three independent biological replicates. Differences were considered as significant at *p* ≤ 0.05.

## 3. Results

### 3.1. Morphological Indicators of Quinoa Sprouts

Growth morphology serves as an intuitive indicator for evaluating the quality of sprouts, making it essential in measuring sprout quality. Generally, it is widely acknowledged that the optimal taste of quinoa sprouts occurs between 12 and 24 h of growth. During this period, the sprouts attain an appropriate biomass and desirable taste. However, if allowed to grow beyond 36 h, the quinoa sprouts become excessively large (Figure 1A–H), resulting in a decline in taste quality.

In our study, as depicted in Figure 1I, we observed that the thousand-sprout weight of quinoa increased with prolonged germination time, indicating the accumulation of biomass after germination. Among the eight varieties analyzed in this experiment, JQ-R1 consistently exhibited the highest thousand-sprout weight, both at the beginning and throughout the germination process. Furthermore, JQ-B2 demonstrated the greatest rate of change in thousand-sprout weight, with increases of 42.1%, 60.3%, and 73.6% after germination for 12, 24, and 36 h, respectively. Notably, the thousand-sprout weight of JQ-W2 displayed rapid initial growth, followed by a slower rate of change in the later stages of germination; with increasing germination time, the diameter of the quinoa seeds (Appendix A) generally exhibits an overall trend of initial enlargement followed by reduction. However, JQ-R2 demonstrates a distinct pattern where the seed diameter does not show an initial enlargement, but rather continuously decreases throughout the germination process; The hypocotyl length of the quinoa seedlings exhibited continuous growth during the germination process, with minimal changes within the first 12 h (Figure 1A–H). However, substantial growth occurred within 24 and 36 h. Specifically, JQ-W2 displayed the longest hypocotyls at 12 and 24 h, measuring 9.610 mm and 18.480 mm, respectively.

### 3.2. Soluble Protein, Soluble Sugar and Free Amino Acid

Soluble protein, soluble sugars, and free amino acids are vital basic nutritional parameters in quinoa, directly absorbed by the human body and crucial for maintaining normal physiological functions. Overall, the soluble protein content in quinoa sprouts exhibits a declining trend during the germination process (Figure 2A) and red varieties display markedly reduced soluble protein content. Specifically, the varieties JQ-R2 and JQ-B2 demonstrate a gradual and gentle reduction, while JQ-W3 does not exhibit a significant difference compared to the baseline at 36 h. Similar trends are observed in the variations in soluble sugars and free amino acids in certain varieties such as W1 and W2 (Figure 2B,C) and, in both cases, red varieties exhibit a significant overall increase at 36 h. Soluble sugar content shows a significant rise at 24 or 36 h compared to 12 h. Regarding free amino acid content (Figure 2C), JQ-W3 experiences a gradual decrease after germination, while JQ-W1, JQ-R1, JQ-R2, and JQ-B2 show a decline, followed by an increase during germination. JQ-W2, JQ-W4, and JQ-B1 initially increase in free amino acid content, peak at 24 h of germination, and subsequently undergo a significant decline.

### 3.3. Hydrolyzed Amino Acid and Gamma-Aminobutyric Acid (GABA)

Hydrolyzed amino acids, in contrast to free amino acids within the plant cells, predominantly exist as peptide chains or proteins, serving as vital nutritional sources. As shown in Figure 3A, with the progression of time, a majority of the hydrolyzed amino acids in quinoa sprouts exhibit an upward trend in their concentrations, including leucine, methionine, and proline. Notably, in the case of JQ-W3, the levels of valine and tryptophan increased significantly by 35.93% and 37.79%, respectively, after 36 h of germination. The content of leucine in JQ-W3 was 1.68-fold higher compared to that of eggs, while tryptophan content showed a 1.87-fold increase after 24 h of germination [20].

Gamma-aminobutyric acid (GABA), a non-protein amino acid, acts as a crucial inhibitory neurotransmitter in the mammalian central nervous system, regulating emotions and seizures. Figure 3B demonstrates a continuous increase in GABA content during the germination of various quinoa sprout varieties. Evidently, there was a remarkable temporal augmentation in the levels of GABA, with a culmination observed at the 36 h mark. Notably, JQ-R2 consistently maintains high GABA content throughout all time periods. Significant variations are observed in JQ-W3, JQ-W1, and JQ-W4, with JQ-W3 showing the most substantial increase, reaching 6.11-fold and 8.26-fold higher GABA content at 24 and 36 h, respectively, compared to 0 h.

### 3.4. Vitamin Analysis

Vitamins are vital for physiological functions. B-group vitamins must be consumed daily as deficiency can lead to skin inflammation and metabolic disorders. Vitamin C, found in plants, protects against scurvy and oxidative stress. The study investigated the levels of B-group vitamins in quinoa sprouts, revealing a consistent upward trend during germination (Figure 4A). Variety JQ-R2 exhibited the highest VB1 content (Figure 4B), reaching 0.892 mg/100 g DW at 24 h and increasing to 1.247 mg/100 g DW at 36 h. VB2 content showed significant variation in JQ-R2 and JQ-W4, with a 4–5-fold increase after 36 h (Figure 4C). VB6 content generally followed the trends of VB1 and VB2, except for JQ-W3 and JQ-B2, which lacked detectable VB6 at certain time points (Figure 4D). Vitamin C (VC) content paralleled the B-group vitamins, increasing with prolonged germination, except for JQ-R2 and JQ-B2 (Figure 4E), where VC decreased significantly from 45.600 mg/100 g DW and 40.900 mg/100 g DW at 0 h to 25.500 mg/100 g DW and 23.430 mg/100 g DW at 36 h, respectively. These findings highlight the variability in vitamin composition and dynamics during quinoa sprout germination.

### 3.5. Total Phenol (TPC) and Total Flavonids (TFC)

Phenolic compounds are vital nutrients with metabolic promotion and antioxidant properties. The phenolic content in quinoa sprouts affects their antioxidant activity, nutrition, and sensory characteristics. Appendix A shows a steady increase in total phenolic content during quinoa sprout germination. After 12 h of germination, some varieties exhibit significant changes, while others remain unchanged compared to non-germinated seeds (e.g., JQ-W3, JQ-W4, and JQ-R1). However, after 36 h, total phenolic content reaches higher levels, such as 19.97 mg/100 g DW for JQ-B2. In contrast, white varieties like JQ-W2 and JQ-W3 have lower contents (3.26 mg/100 g DW and 4.26 mg/100 g DW, respectively). The results indicate a correlation between total phenolic content and seed coat color, with higher levels in black and red varieties.

In this experiment, the total flavonoid content was measured to assess differences among quinoa varieties during growth (Appendix A), which revealed a significant increase in total flavonoid content at 24 h compared to 0 h. JQ-W3 consistently exhibited the highest total flavonoid content at all growth time points. At 24 and 36 h of germination, JQ-W3 contained 1181.82 and 1517.27 g/100 g DW, respectively. In contrast, JQ-R1 consistently had lower total flavonoid content, with 208.49 and 325.76 g/100 g DW at the same time points. Varieties with initially lower total flavonoid content maintained lower levels throughout the sprouting process.

In summary, the content of phenolic and flavonoid compounds in quinoa sprouts increases with germination time. Varieties with darker seed coat colors tend to have higher levels of phenolic compounds. The initial content of these compounds in the seeds plays a significant role in determining their levels during germination.

### 3.6. Saponins Analysis

Saponins, bioactive compounds with antioxidant properties, are often disliked by consumers due to their bitter taste. Therefore, determining the optimal saponin content in quinoa sprouts is crucial to enhance their palatability. The measurement results of saponin content in Appendix A demonstrate a general decreasing trend, followed by an increasing trend during the growth process of quinoa sprouts. In the early stages of germination (0–12 h), there is a significant rapid decline in saponin content, reaching its lowest level. However, by 24 h of germination, there is a significant increase in saponin content compared to 12 h, and the saponin content in most varieties at 24 h is comparable to that of ungerminated quinoa seeds. As the quinoa sprouts continue to germinate until 36 h, there is a further significant increase in saponin content, even surpassing that of ungerminated quinoa seeds. JQ-B2 exhibited consistently high saponin content throughout germination, reaching 1.79% at 36 h, potentially affecting the sensory experience.

### 3.7. Antioxidant Activities

Quinoa sprouts exhibited a progressive and significant increase in antioxidant activity during the germination process, surpassing the antioxidant potential of the seeds. Regarding DPPH, JQ-W1 showed minimal changes during sprouting, with only a 19.28% increase at 36 h, while JQ-W3 increased by 164.32%. For FRAP, JQ-W2 had the highest activity at 24 h (2.93 mmol Fe^2+^ equivalents/100 g), while JQ-B2 showed the highest activity at 36 h (3.95 mmol Fe^2+^ equivalents/100 g), 1.78 times higher than at 0 h. In terms of ABTS, both JQ-W3 and JQ-W4 maintained consistently high activity levels, ranging from 9832.53 to 11,989.46 mM TEAC/g DW. To comprehensively compare the antioxidant capacity of different varieties at different time points, the antioxidant potency composite (APC) was calculated in this study. As indicated in Table 2, JQ-B1 consistently exhibited the highest antioxidant activity, while JQ-W1 demonstrated the lowest activity. Moreover, a positive correlation (r = 0.643) was observed between the total APC and soluble protein content (Appendix A), suggesting the presence of specific antioxidant components within soluble proteins that contribute to the overall antioxidant activity.

### 3.8. Principal Component Analysis

Principal component analysis (PCA) was used to analyze quinoa sprouts from eight varieties at different time points, considering nutritional components, flavonoids, phenolics, antioxidant activity, and saponins. Soluble protein and sugar had the most significant impact on the sprout characteristics, explaining 57.9% of the total variance (PC1: 39.9%, PC2: 18%). Dimension 1 (Soluble protein) and Dimension 2 (Soluble sugar) are spatially close to perpendicular, indicating their relative independence (r = −0.15). GABA, soluble proteins, and saponins contributed the most to Dimensions 1 and 2. Increasing processing time led to higher levels of amino acids and phenols (PC1), while soluble proteins and antioxidant capacity decreased. In the biplot (Figure 5E), from left to right, with increasing time, most nutrients such as GABA, vitamins, soluble sugars, and saponins also increase. There were significant differences between the 0 h and 24 h/36 h treatments. However, saponin content also increased after 24 h. Thus, the quinoa sprouts harvested and processed at 24 h had higher nutritional value and lower saponin content.

## 4. Discussion

During the growth process of quinoa sprouts, the progressive increase in biomass and diameter signifies the accumulation of nutrient reserves and enhanced endosperm content [21], a phenomenon which aligns with observations in sprouts of other plant species, such as wheat and soybeans [22,23]. Subsequently, the reduction in seed diameter likely reflects the mobilization of stored nutrients to facilitate the elongation and development of the hypocotyl.

Soluble proteins, soluble sugars, and free amino acids not only sustain human metabolic processes but also serve as foundational nutritional substrates in plants [24,25]. Soluble sugars, as a whole, attain their peak levels during germination at approximately 24 to 36 h, displaying varietal disparities. A reciprocal equilibrium exists between soluble proteins and free amino acids, where species exhibiting declining protein content during germination showcase a concomitant increase in free amino acid levels. This phenomenon exhibits a similar inclination observed in the germination of other botanical sprouts [26], underscoring the interdependence of these nutritive constituents in plant physiology.

Quinoa sprouts exhibit the synthesis and accumulation of amino acids, including Leu and Asp, during growth, consistent with findings from Fouad et al. in their research on mung bean germination [27]. Quinoa is recognized for its high content of essential amino acids like lysine and leucine, surpassing that of cereal crops [28]. This study reveals elevated levels of leucine, glycine, and lysine in quinoa sprouts. Compared to spinach [29] grown in the field and greenhouse (aspartic acid content: 95.3 and 173.1 mg/100 g, respectively), quinoa sprouts offer a significant source of dietary protein with higher concentrations of essential amino acids. Variations in quinoa sprouts’ amino acid content and composition may arise from factors such as variety, environment, and interactions [30].

GABA, a non-proteinogenic amino acid found universally, has gained recognition as a pivotal modulator of plant growth and development under varying environmental conditions [31]. The present study revealed a consistent elevation in GABA content during the germination of quinoa sprouts, corroborating the findings of Jiao et al. [32]. The modulation of GABA synthesis and degradation involves intricate regulatory mechanisms influenced by genetic variations and metabolic control [33,34]. Consequently, the observed variations in GABA content among different quinoa varieties in this experiment can be attributed to the following underlying factors.

Thiamine (VB1) functions as a cofactor for enzymes involved in bioenergetics, amino acid metabolism, and carbohydrate conversion [35]. The present study observed a progressive increase in the content of vitamin B1 during germination, consistent with findings by Coello et al. [36]. The VB1 content in non-germinated seeds ranged from 0.313 to 0.617 mg/100 g DW, in accordance with findings by Miranda et al. [37]. Riboflavin (VB2) is an essential component for fundamental metabolism and serves as a precursor to coenzymes FAD and FMN ADDIN [38]. In this experiment, a significant variation in VB2 content was observed in JQ-R2 and JQ-W4, with a 4–5-fold increase after 36 h. This phenomenon aligns with related findings in germinated oats, wheat, and rye [39], all exhibiting higher riboflavin levels compared to non-germinated grains. Pyridoxine (VB6) serves as a coenzyme in the biosynthesis of niacin and is formulated in medications for treating rough skin [40]. In this study, the content of VB6 in germinated amaranth seeds ranged from 0.187 to 1.481 mg/100 g DW, significantly higher than that of the wheat sprouts, but similar to that of the radish sprouts [41]. Vitamin C, a vital nutritional component in vegetables, exhibits remarkable antioxidant activity. After 36 h of germination, most varieties demonstrate elevated vitamin C content, displaying significant inter-variety variations ranging from 23.43 to 59.7 mg/100 g DW. This aligns with the comparable vitamin C levels observed in 2–7-day-old quinoa sprouts [42].

Polyphenolic compounds, as secondary metabolites in plants, undergo changes during germination, leading to increased content and enhanced antioxidant capacity [43]. In our study, TPC levels progressively increased with germination time, ranging from 4.26 to 19.97 mg/g DW at 36 h. This differs from Petrucci’s findings (19.62–28.58 mg/g DW at 72 h), likely due to variations in germination time and solvent selection during soaking [44]. Quinoa seed color is modulated by a spectrum of compounds and their interplay. The black variety JQ-B2 exhibited the highest TPC, while the white variety JQ-B2 had the lowest, consistent with Tang’s research [45]. TFC levels significantly increased in sprouts compared to 0 h, resembling Petrucci’s findings [44]. Only JQ-R1’s TFC content at 24 h matched Niu et al.’s report [46], while other varieties displayed higher levels. Overall, black quinoa sprouts exhibit elevated phenolic content, contrasting with the white varieties rich in flavonoids. This study posits that antioxidative agents likely govern the black varieties, while the white variants may necessitate heightened flavonoids for tailored metabolic and growth requisites.

Saponins possess antioxidant activity but are rejected by consumers due to their bitter taste [47]. Our study found an initial decline in saponin content during 0–12 h, similar to researcher Marco’s findings [48]. However, except for JQ-B1, the other varieties reached their initial levels at 24 h and peaked at 36 h. Aside from the varieties JQ-W3, JQ-R1, JQ-R2 and JQ-B2 at 24 h, saponin content in the other varieties exceeded 5 mg/g DW, making them unsuitable as commercial grains [49]. However, these other varieties can be used as natural surfactants and preservatives in food [48].

The antioxidant activity of quinoa sprouts increased during the germination process. In this experiment, antioxidant activity in the quinoa sprouts correlated strongly with soluble proteins (r = 0.64) and free amino acids (r = −0.52), reflecting their antioxidant potential. At 36 h of germination, JQ-B1 exhibited the highest DPPH radical scavenging capacity, 1.69 times that of 0 h. This phenomenon is similar to the observed increment in radish and sunflower sprouts but higher than in broccoli sprouts [43]. Regarding FRAP, its content, ranging from 2.02 to 3.95 mmol Fe^2+^ equivalents/100 g, exhibited a substantial elevation when compared to that of mung bean sprouts [50]. APC showed the strongest correlation with soluble proteins (r = 0.643), indicating their potential antioxidative properties. This relationship suggests that soluble proteins may directly neutralize oxygen radicals or indirectly modulate antioxidant processes. Furthermore, the stability and integrity of cellular proteins are crucial for normal cellular functions and antioxidant defenses.

## 5. Conclusions

Quinoa sprouts surpass quinoa seeds in terms of nutritional value, especially with higher levels of lysine, vitamin B, and antioxidants, making them stand out among common cereal grains. Using principal component analysis, this study visualized the relationships among nutrients, antioxidant activity, and saponin content in quinoa sprouts at different time points. The black quinoa JQ-B1 and white quinoa JQ-W3, sprouted for 24 h, were selected for their higher amino acid, total flavonoid, and vitamin content, along with lower saponin levels. Additionally, seed coat color showed a positive correlation with total phenolic content, with the black varieties exhibiting higher phenolic compound levels compared to the red and white varieties. Overall, this research provides valuable insights into the nutritional composition of quinoa sprouts during growth and offers a visual method for evaluating different varieties at different time points, supporting the integration of quinoa sprouts into the progressive development of the food industry.

## Figures and Tables

**Figure 1 foods-13-02513-f001:**
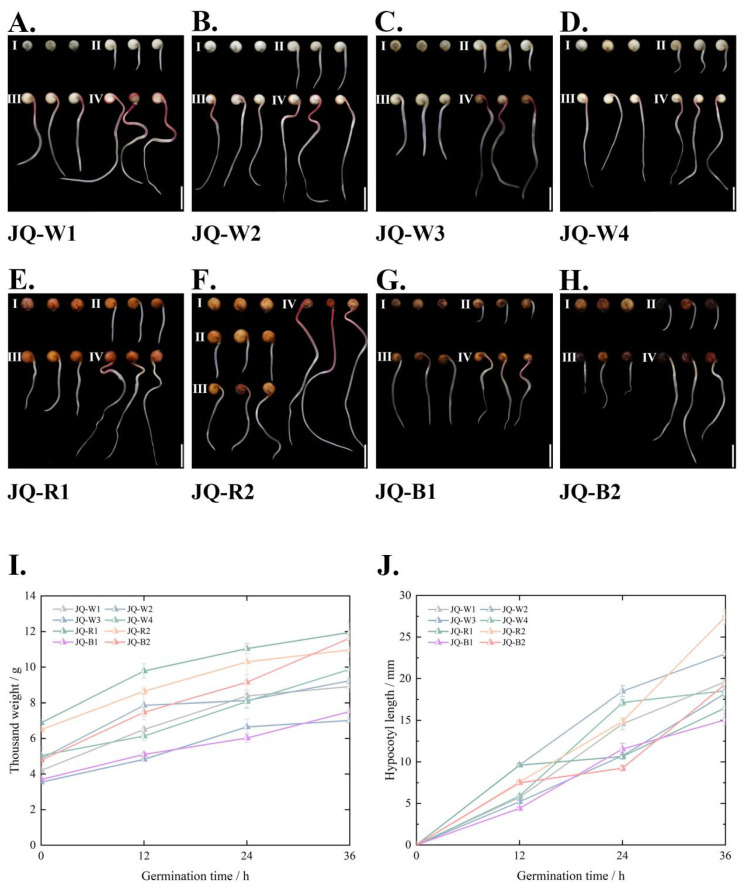
The growth phenotype, thousand-sprout weight, and hypocotyl length of quinoa from seeds to sprouts at different germination time points. Figure (**A**–**H**) represent different quinoa varieties, and figure legends, Bar = 1 cm. I–IV, respectively, represent germination at 0, 12, 24, 36 h. The data at each time point in the figure are mean ± SD, n = 3. Figure (**I**,**J**) represent thousand weight and hypocotyl length, respectively.

**Figure 2 foods-13-02513-f002:**
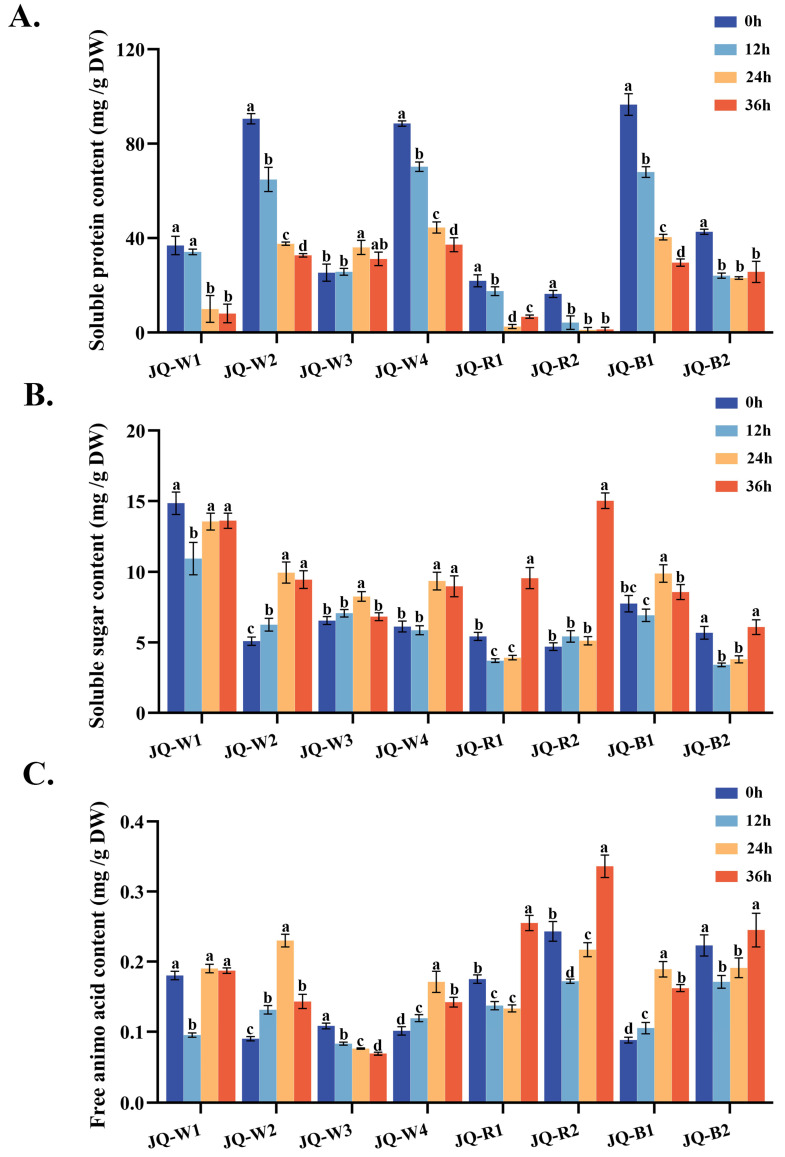
Soluble protein content (**A**), soluble sugar content (**B**) and free amino acid content (**C**) of quinoa at different germination time points. Lowercase letters denote notable variations within the same quinoa variety across distinct growth stages. The data at each time point in the figure are mean ± SD, n = 3.

**Figure 3 foods-13-02513-f003:**
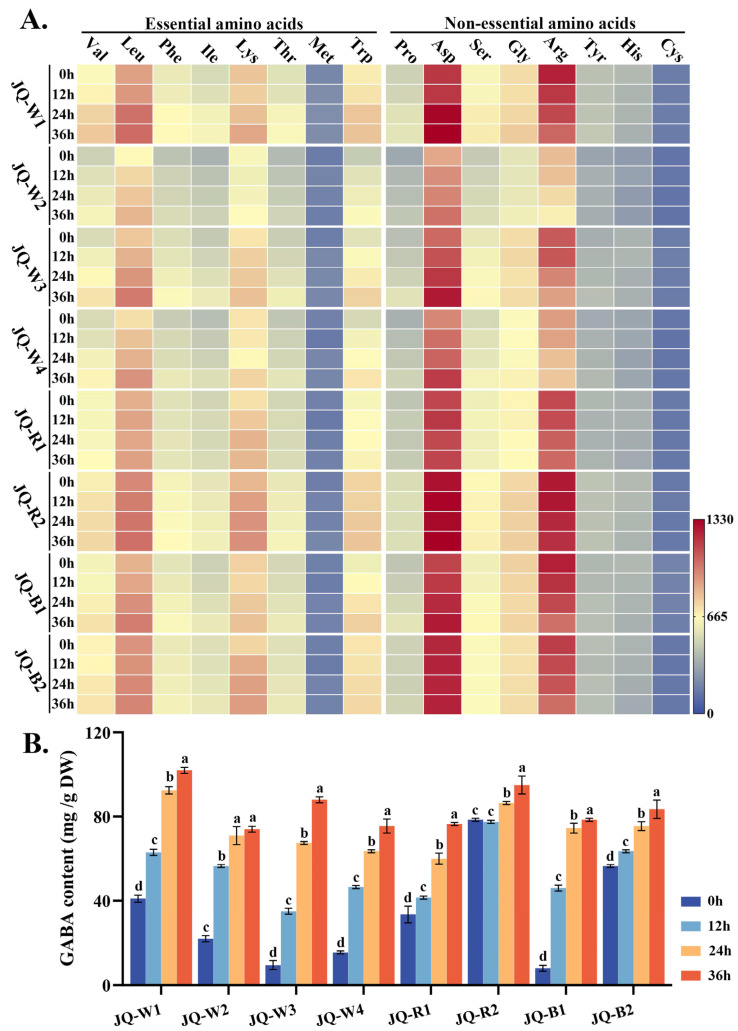
Hydrolyzed amino acid (**A**) and GABA content (**B**) of quinoa at different germination time points. In the form of heat map, the variation trend of various amino acid contents is indicated. Blue indicates low content, while red indicates high content. Lowercase letters serve as indicators of significant phenotypic variations within the same quinoa variety across distinct growth stages. The data at each time point in the figure are mean ± SD, n = 3.

**Figure 4 foods-13-02513-f004:**
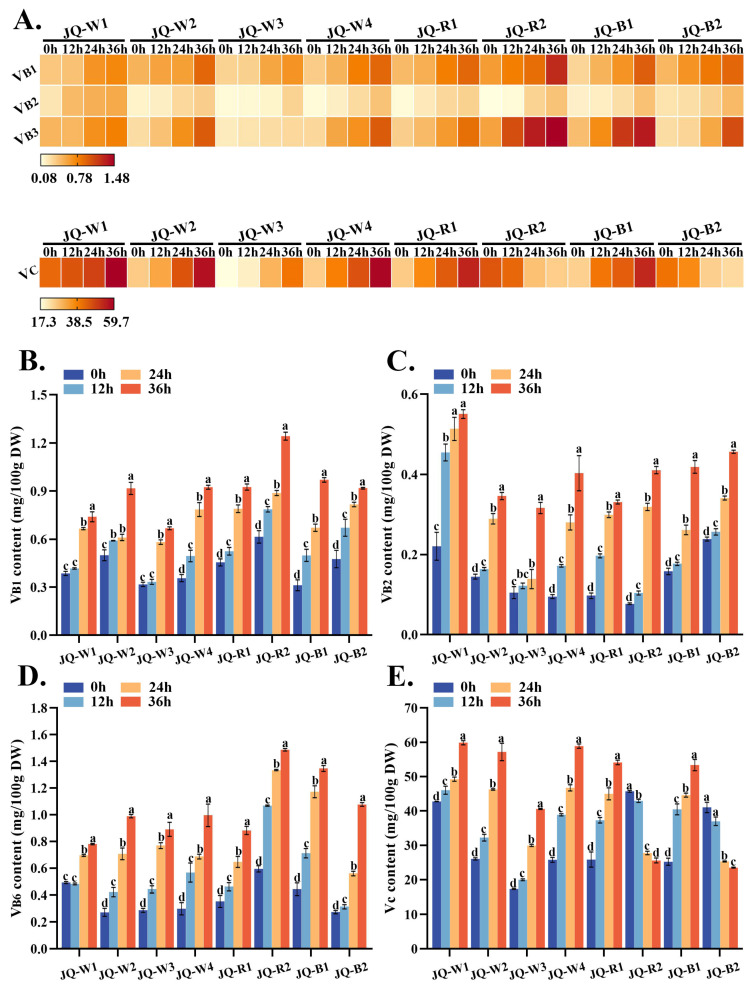
In the heat map (**A**), the variation trend of various vitamin contents is indicated; increased red color indicates greater substance concentration, while whiter color indicates lower content. (**B**–**E**) represents VB1, VB2, VB6, and VC content (mg/100 g DW) of quinoa sprouts at different time points, respectively. Lowercase letters serve as indicators of significant phenotypic variations within the same quinoa variety across distinct growth stages. The data at each time point in the table are mean ± SD, n = 3.

**Figure 5 foods-13-02513-f005:**
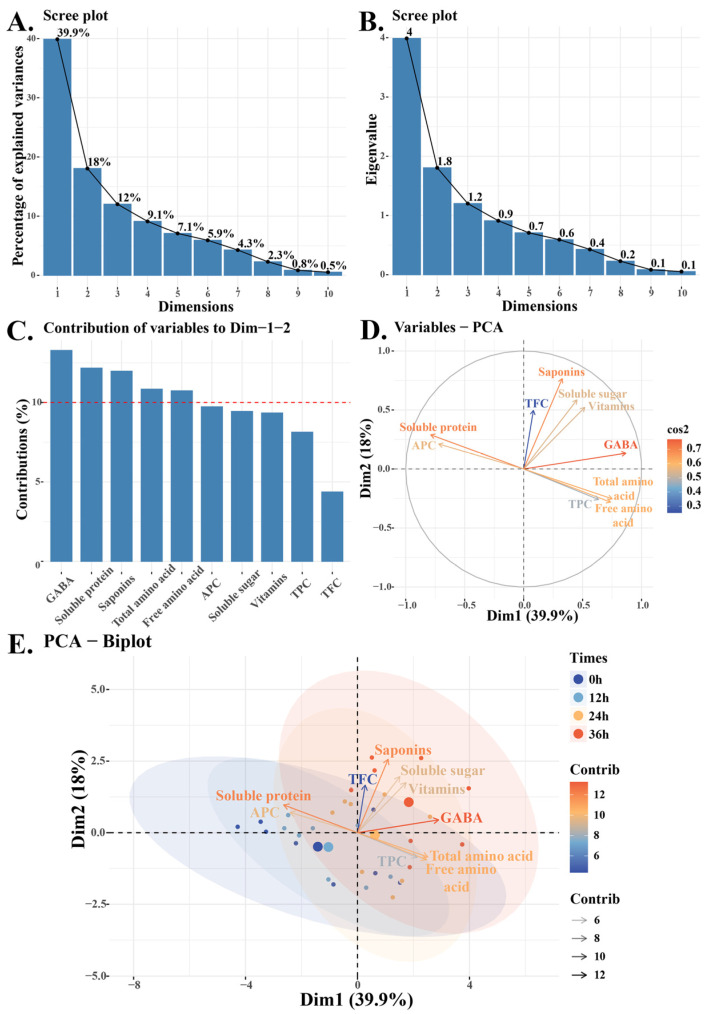
The scree plot (**A**,**B**), contribution plot (**C**), variables PCA (**D**) and biplot (**E**) in principal component analysis (PCA) is utilized to assess the nutritional content and antioxidant activity of different varieties of quinoa sprouts at different time points. Dimension 1 (Dim1) means soluble protein, and Dimension 2 (Dim1) means soluble sugar. The red dashed line represents the expected average contribution.

**Table 1 foods-13-02513-t001:** Basic information about eight varieties of quinoa seeds.

Number	Seed Coat Color	Origin	Common Name
JQ-W1	white	Bolivia	DB-2
JQ-W2	white	Bolivia	WXY-9-4
JQ-W3	white	Chile	Faro
JQ-W4	white	USA	Brightest Brilliant
JQ-R1	red	Bolivia	WXY-1-Y2
JQ-R2	red	Bolivia	WXY-12-1
JQ-B1	black	Argentina	JQ-B-2
JQ-B2	black	Bolivia	WXY-13-4

**Table 2 foods-13-02513-t002:** Antioxidant activity of different quinoa sprouts at different time points.

Time	Sample	DPPH (%)	FRAP (mmol Fe^2+^ Equivalents/100 g)	ABTS (mM TEAC/g DW)	APC	Rank
0 h	JQ-W1	6.95 ± 1.27 c	2.10 ± 0.22	7012.34 ± 239.45 f	73.19	7
	JQ-W2	8.99 ± 0.45 b	2.33 ± 0.21	8912.79 ± 79.35 c	88.81	2
	JQ-W3	6.39 ± 0.72 c	2.02 ± 0.14	9832.53 ± 191.55 b	79.47	4
	JQ-W4	5.62 ± 0.55 c	2.28 ± 0.13	10,266.53 ± 87.24 a	82.31	3
	JQ-R1	6.51 ± 0.98 c	2.20 ± 0.26	8097.56 ± 92.43 d	76.74	5
	JQ-R2	6.39 ± 1.98 c	2.15 ± 0.37	8135.69 ± 63.76 d	75.95	6
	JQ-B1	11.09 ± 0.33 a	2.37 ± 0.37	7567.90 ± 106.35 e	91.24	1
	JQ-B2	5.62 ± 0.55 c	2.22 ± 0.17	7478.09 ± 21.64 e	72.48	8
12 h	JQ-W1	7.39 ± 0.72 b	2.47 ± 0.12	7095.52 ± 56.43 g	76.52	7
	JQ-W2	9.52 ± 0.26 a	2.43 ± 0.11	9435.75 ± 75.36 c **	89.91	2
	JQ-W3	7.96 ± 0.91 b	2.51 ± 0.16 *	9962.70 ± 74.97 b	87.77	4
	JQ-W4	6.93 ± 1.06 b	2.62 ± 0.16 *	10,747.80 ± 59.34 a **	88.27	3
	JQ-R1	8.14 ± 0.82 b	2.52 ± 0.36	8126.70 ± 173.85 d	82.76	5
	JQ-R2	6.97 ± 0.46 b	2.54 ± 0.21	8263.46 ± 38.64 d *	79.73	6
	JQ-B1	10.7 ± 0.57 a	2.59 ± 0.22	7845.64 ± 83.47 e *	90.62	1
	JQ-B2	6.93 ± 1.06 b	2.41 ± 0.25	7643.64 ± 95.37 f *	75.97	8
24 h	JQ-W1	7.39 ± 0.65 c	2.89 ± 0.16 **	7265.76 ± 209.24 f	70.98	7
	JQ-W2	10.00 ± 0.43 b *	2.93 ± 0.18 *	9523.64 ± 46.67 c ***	83.94	2
	JQ-W3	8.82 ± 0.83 b *	2.68 ± 0.27 *	10,678.58 ± 49.35 b **	81.55	4
	JQ-W4	7.15 ± 0.12 c **	2.83 ± 0.27 *	11,909.36 ± 135.45 a ***	82.71	3
	JQ-R1	9.66 ± 1.16 b *	2.64 ± 0.21	8167.68 ± 306.23 e	76.08	6
	JQ-R2	10.14 ± 1.49 b	2.70 ± 0.11	8533.63 ± 30.34 d **	78.94	5
	JQ-B1	13.91 ± 0.6 a **	2.76 ± 0.14	8054.74 ± 57.73 e **	87.39	1
	JQ-B2	7.15 ± 0.12 c **	2.87 ± 0.37 *	7535.26 ± 286.45 f	70.93	8
36 h	JQ-W1	8.29 ± 0.47 d	2.99 ± 0.26 b *	7534.76 ± 57.99 f *	60.83	8
	JQ-W2	13.47 ± 1.15 c **	3.02 ± 0.27 b *	9529.47 ± 27.35 c ***	75.87	6
	JQ-W3	16.89 ± 0.38 b ***	3.22 ± 0.38 b **	11,098.87 ± 290.46 b **	87.94	1
	JQ-W4	12.72 ± 0.09 c ***	3.33 ± 0.40 b *	11,989.46 ± 104.46 a ***	83.94	2
	JQ-R1	12.72 ± 0.15 c ***	3.59 ± 0.13 ab **	8556.48 ± 83.46 d **	76.67	5
	JQ-R2	13.95 ± 2.07 c **	3.13 ± 0.19 b *	8593.25 ± 57.46 d **	75.00	7
	JQ-B1	18.77 ± 1.62 a **	3.06 ± 0.17 b *	8154.64 ± 140.34 e **	81.75	3
	JQ-B2	12.72 ± 0.09 c ***	3.95 ± 0.41 a **	7964.24 ± 46.80 e ***	78.06	4

Lowercase letters signify significance among different varieties simultaneously, with *p* < 0.05. The * indicates significance within the same variety at different times, using the 0 h value as the reference point, where * denotes *p* < 0.05, ** denotes *p* < 0.01, and *** denotes *p* < 0.001. The data at each time point in the table are mean ± SD, n = 3.

## Data Availability

The original contributions presented in the study are included in the article/Appendix A, further inquiries can be directed to the corresponding author.

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
