# Peer review of "Comparative Evaluation of Chemical Composition and Nutritional Characteristics in Various Quinoa Sprout Varieties: The Superiority of 24-Hour Germination"

_foods, 2024, doi:10.3390/foods13162513_

Round 1

Reviewer 1 Report

Comments and Suggestions for Authors

See the attachment

Reviewer 2 Report

Comments and Suggestions for Authors

Comments to authors

The article titled “Comparative Evaluation of Chemical Composition and Nutritional Characteristics in Various Quinoa Sprout Varieties: The Superiority of 24-hour Germination” has been revised and I would like to draw your attention to some important points. The necessary revisions are as follows:

Lines 38-41. In relation to “[...] Studies have found that quinoa contains a large number of proteins and biologically available essential amino acids, unsaturated lipids, complex carbohydrates, dietary fiber and other useful bioactive compounds. […]” insert the references of these studies.

 Line 44. Instead of writing “[...] In recent years [...]” write something like “In addition…”.

Line 64. Regarding “[...] but the quinoa sprouts’ cultivation, nutritional value, processing and utilization have not been fully studied and discussed […] ‘’. It is not clear whether this is an argument of the authors or of the references [9-11]. If it is the authors’ opinion based on the analysis of the cited references, I recommend that the authors review by changing the position of the references to line 64 “[...] quinoa seeds [9-11], but the quinoa sprouts’ cultivation, nutritional value, processing and utilization have not been fully studied and discussed […] ‘’.

Lines 67-68. “[…] Thus, this paper will study the changes of various nutritional components during the growth and development of quinoa sprouts. […]”. Only the nutritional components? And the phenolic compounds, saponin and antioxidant activities were also not studied? I recommend that the authors revise the text to make clearer the purpose of the study and which knowledge gap it intends to fill.

Lines 72-75. The authors state that “[...] At present, there are few reports about quinoa sprouts worldwide…[…]”, however, the literature presents a large number of scientific works published in different countries and in different languages. Review the text by replacing it with a sentence that shows the contribution of the study, such as “This study provides a reference...

Line 84. Adds information (model, brand, country of manufacture) about the equipment. Report the relative humidity during the germination process.

Lines 95-139: In the analyses of soluble sugar, soluble protein and free amino acid, amino acid and GABA, vitamin content and saponin the terms powdered samples, dry sample powder and dehydrated sample are used. While, in the analyses of total phenols content and total flavonoids content and antioxidant activities, the terms quinoa samples and samples are used, respectively, to describe the sample used in the analyses. Before the analyses, were the samples prepared in different ways? Therefore, it would be interesting to include how the samples were prepared before the analyses. This is important in case another research group wants to reproduce the results.

It would also be interesting to perform analysis of variance and show whether there are statistical differences revealed between the varieties demonstrated and the germination times in Figure 1I-J and Figure S1.

Line 199. Figure 1H?

I wonder why the bottom of the error bars are not shown in Figures 2, 3, 4 and 5. I also wonder why the results are expressed on a fresh basis instead of a dry basis. If possible, use the same basis for all results, fresh or dry basis.

Present Figure 5 after it is first mentioned in the text.

In Table 1, indicate whether or not there was a significant difference between the different treatments (varieties and germination times). Use lowercase or uppercase letters to indicate statistically significant variations within the same quinoa variety at different growth stages and significant variations between quinoa varieties at the same growth stages.

Line 364. Unless it is a rule of the journal, where it says “[...] Fouad A A [...]” write “Fouad et al.”. The same goes for the other references cited in the text (for example, line 375: “Jiao's” as “Jiao et al.; line 383: “Coello” as “Coello et al.”; line 384: “Miranda” as “Miranda et al.”, etc.). Review the entire manuscript and standardize the format of the references cited in the text.

Throughout the manuscript, consider replacing the term “cultivation” with “germination”, except, of course, if it is related to agronomic aspects.

 Review the formatting of the list of references.

Round 2

Reviewer 1 Report

Comments and Suggestions for Authors

The authors could additionally improved the section Material and Methods since they did not satisfactorily responded to some queries:

- “standard amino acid solutions” should be defined

- HPLC conditions for GABA measurements were not provided.

- Expression of results for total phenolics and flavonoids was not provided.
